# Combined CT Coronary Artery Assessment and TAVI Planning

**DOI:** 10.3390/diagnostics13071327

**Published:** 2023-04-03

**Authors:** Matthias Renker, U. Joseph Schoepf, Won Keun Kim

**Affiliations:** 1Department of Cardiology, Campus Kerckhoff of the Justus Liebig University Giessen, 61231 Bad Nauheim, Germany; 2Department of Cardiac Surgery, Campus Kerckhoff of the Justus Liebig University Giessen, 61231 Bad Nauheim, Germany; 3German Centre for Cardiovascular Research (DZHK), Partner Site Rhine-Main, 61231 Bad Nauheim, Germany; 4Heart & Vascular Center, Medical University of South Carolina, Charleston, SC 29425, USA; 5Department of Cardiology, Justus Liebig University Giessen, 35392 Giessen, Germany

**Keywords:** aortic stenosis, chronic coronary syndrome, coronary artery disease, computed tomography, fractional flow reserve, transcatheter aortic valve implantation

## Abstract

Computed tomography angiography (CTA) of the aorta and the iliofemoral arteries is crucial for preprocedural planning of transcatheter aortic valve implantation (TAVI) in patients with severe aortic stenosis (AS), because it provides details on a variety of aspects required for heart team decision-making. In addition to providing relevant diagnostic information on the degree of aortic valve calcification, CTA allows for a customized choice of the transcatheter heart valve system and the TAVI access route. Furthermore, current guidelines recommend the exclusion of relevant coronary artery disease (CAD) prior to TAVI. The feasibility of coronary artery assessment with CTA in patients scheduled for TAVI has been established previously, and accumulating data support its value. In addition, fractional flow reserve determined from CTA (CT–FFR) and machine learning-based CT–FFR were recently shown to improve its diagnostic yield for this purpose. However, the utilization of CTA for coronary artery evaluation remains limited in this specific population of patients due to the relatively high risk of CAD coexistence with severe AS. Therefore, the current diagnostic work-up prior to TAVI routinely includes invasive catheter coronary angiography at most centers. In this article, the authors address technological prerequisites and CT protocol considerations, discuss pitfalls, review the current literature regarding combined CTA coronary artery assessment and preprocedural TAVI evaluation, and provide an overview of unanswered questions and future research goals within the field.

## 1. Introduction

A significant proportion of patients with severe aortic stenosis (AS) present with concomitant coronary artery disease (CAD): approximately 15–80% of the patients evaluated for transcatheter aortic valve implantation (TAVI) have coexistent CAD, a figure that increases with age in parallel with operative risk factors [1,2]. The consideration of CAD in the context of TAVI is justified by the conception that these patients have an elevated risk of procedural complications and worse outcomes. It is argued that revascularization before TAVI may prevent myocardial ischemia during phases of hypotension and/or rapid pacing [3]. For this reason, current guidelines primarily advocate invasive coronary angiography (ICA) as the standard method for coronary artery assessment in all TAVI candidates [4,5]. The indication for revascularization prior to TAVI remains a focus of ongoing research. Available study data in this context is limited. The ACTIVATION trial by Patterson et al. [6] found similar rates of death and rehospitalization at one year when comparing percutaneous coronary intervention with no revascularization in patients prior to TAVI, although the prespecified noninferiority margin was not met. Because the inclusion of patients relied on stenosis >70% alone, functional lesion assessment could be a more appropriate strategy for revascularization prior to TAVI. Nevertheless, guidelines suggest excluding >70% diameter stenosis of the proximal coronary artery segments [4]. For this reason, results of the ongoing trials NOTION-3 (ClinicalTrials.gov Identifier: NCT03058627), TAVI PCI (ClinicalTrials.gov Identifier: NCT04310046) and COMPLETE TAVR (ClinicalTrials.gov Identifier: NCT04634240) are eagerly awaited.

Due to the need for preprocedural diagnostic information, computed tomography angiography (CTA) of the aorta and the iliofemoral arteries has become the mainstay of TAVI planning. It provides details on a variety of aspects required for heart team decision-making. In addition to supplying relevant diagnostic information on the degree of aortic valve calcification, CTA allows for a customized choice of the transcatheter heart valve and the TAVI access route. Electrocardiogram (ECG) synchronization and narrow-width detector collimation are prerequisites for an accurate characterization of the aortic annulus and for prosthesis selection. At the same time, these CT system requirements are crucial for coronary artery image acquisition. The feasibility of coronary artery assessment with CTA in a subset of patients scheduled for TAVI has been established previously and based on further development in the field of CT technology, with improved spatial and temporal resolution, its potential value has been suggested [7,8,9,10,11,12,13,14,15,16,17,18,19,20,21,22,23]. Only recently, the addition of computational fluid dynamic-based and machine learning (ML)-based solutions for fractional flow reserve from CTA (CT–FFR) showed the potential to improve the diagnostic performance of CTA before TAVI [24,25,26,27,28]. However, the utilization of CTA for coronary artery evaluation remains limited in this specific population of patients due to the relatively high risk of CAD concomitant with severe AS and the present guideline statement.

The present review addresses technological prerequisites as well as CT protocol considerations, considers pitfalls, reviews the current literature regarding an integrated coronary artery assessment during routine preprocedural CTA for TAVI, and provides an overview of unresolved issues and perspectives for future research within this field of interest.

## 2. Technological Prerequisites

General recommendations exist for CTA in the setting of both assessment prior to TAVI and coronary artery evaluation [29,30]; however, these documents do not provide an orientation towards a certain CT scanner type. From the authors’ experience, 64-detector row technology is the minimum requirement. This is in line with the large body of literature showing improved diagnostic performance of advanced 64-section CT scanner generations for coronary CTA in patients without AS [31]. Moreover, given the complexity of the often multimorbid population of TAVI candidates (e.g., due to the presence of cardiac arrhythmias, chronic kidney disease, increased coronary artery calcification), utilization of the most contemporary CT scanner generation is advisable.

## 3. Prescan Considerations and Patient Preparation

Among patients with severe AS and impaired renal function, prehydration is not recommended due to the increased risk of cardiac decompensation and the absence of evidence that prehydration is beneficial [32,33]. Although patients with chronic kidney disease but with a GFR ≥30 mL/min do not seem to require any prophylactic measures before CTA, those with more severe renal impairment or in case of an emergency setting may benefit from it [34]. Hence, a balanced fluid management with careful prehydration in dehydrated patients, depending on their renal function and setting of hospital admittance is recommended. The amount of contrast agent administered can be determined based on renal function, patient habitus, CT protocol, and scanner generation. The bolus of iodinated contrast agent is injected through an intravenous line of at least 18–20 gauge that is best placed in an antecubital vein. The flow rate of 3.5 to 6 mL/s can be adjusted according to the patient habitus and the iodine concentration of the contrast material used [29].

In the CAST–FFR study by Michail et al. [24], prescan medication was administered in patients with AS in order to optimize image quality of CTA prior to TAVI. Although nitrates are traditionally avoided in patients with AS out of concern for hypotension and circulatory failure, all patients were carefully prehydrated and then received 0.4 mg sublingual nitroglycerin in this study. In addition, beta-blockers and ivabradine were used to achieve a heart rate below 60/min. Notably, the authors report that no patients incurred complications due to the prescan medication or CTA, and that 92% of the study cohort had interpretable CTA data. It is important to acknowledge that these were selected patients who were closely monitored in adherence to strict prescan protocols. Therefore, this approach may not be generally applicable.

## 4. CT Protocol

### 4.1. Calcium Scoring

Before CTA, an initial noncontrast-enhanced calcium scoring examination with standard protocol parameters can be performed. For this purpose, the scan range is chosen from the carina to the apex of the heart using a prospective ECG-triggering technique, 120-kVp tube voltage, diastolic phase data reconstruction, and a slice thickness of 3 mm. From this sequence, the coronary Agatston score as well as the aortic valve calcium score are computed [35,36]. Relevant other calcium measures, such as volume, mass, and distribution, are known but will not be the focus of this article.

Both the coronary Agatston score, and the aortic valve calcium score offer valuable diagnostic and prognostic information with low additional radiation dose. Although these scores have been shown to be available from CTA as well, the noncontrast-enhanced sequence remains the standard of reference [37,38,39]. The coronary Agatston score has been shown to be useful for ruling out obstructive CAD in the CT-evaluation before TAVI and thereby allowing for the reduction of a significant number of ICAs [40]. Additionally, Cartlidge et al. recently assessed the correlation of aortic valve calcific and noncalcific volumes from CTA with AS severity and found that this method may be preferable to noncontrast CT, especially in cases with fibrosis as the main contributor to AS [41].

### 4.2. CTA Acquisition Technique

As summarized by the latest expert consensus document of the Society of Cardiovascular Computed Tomography by Blanke et al. [29] on CT imaging in the context of TAVI, two CTA acquisition strategies are to be considered for preprocedural assessment prior to TAVI:An ECG-synchronized acquisition of the heart including the aortic root that is followed by a non-ECG-synchronized acquisition from the thorax to the pelvis;An ECG-synchronized acquisition of the whole thorax that is followed by a non-ECG-synchronized CTA from the abdomen to the pelvis.

To allow for coronary artery assessment, at least the heart and the aortic root complex should be scanned using ECG synchronization. While ECG synchronization of the entire thorax excludes redundancy of the scanned areas and ensures ECG-synchronized coverage of the coronary artery tree, this approach increases the thoracic scan time and confers a higher radiation dose and a higher risk of breathing artifacts.

Depending on the CT system, prospective ECG-triggering or retrospective ECG-gating acquisition techniques are used in routine practice. An important advantage of retrospective ECG gating with data acquisition throughout the whole cardiac cycle is that the aortic annulus and the coronary arteries can be reconstructed in systole as well as in diastole. Aortic annulus measurements are subject to conformational changes. Systolic annulus measurements are usually larger and determine the choice of the transcatheter heart valve. In contrast, the timing of coronary artery image acquisition is typically facilitated by more tranquil cardiac movement during mid-diastole for lower heart rates. Moreover, in cases with artifacts that are due to factors such as arrhythmias or high heart rates, the retrospective ECG-gating method allows selection of the heart phase with the least artifacts or motion to salvage diagnostic image quality. However, this comes at the cost of a higher radiation dose in comparison with prospective ECG-triggering. Figure 1 illustrates excellent image quality of the coronary arteries using the retrospective ECG-gating technique for CT evaluation of a patient referred for TAVI.

The two-part CTA protocol generally involves a single bolus of contrast agent. It is recommended that CTA acquisition be started automatically via bolus tracking with a set delay after the signal attenuation in the ascending aorta has reached the predefined level of 120 HU. Differences between the CTA scan parameters obviously apply that are dependent on the CT system being used.

## 5. Data Reconstruction and Image Interpretation

Iterative reconstruction techniques have been shown to improve image quality and/or reduce radiation dose in comparison with traditional filtered backprojection. Therefore, iterative reconstruction has become the standard of care and should be used [42]. Furthermore, it has been shown that high-spatial-resolution convolution kernels are useful for coronary artery assessment, especially in the presence of extensive coronary atherosclerosis and calcifications [43]. For this reason, it is advisable to implement not only “smooth” vascular convolution kernels but also a “sharp” high-spatial-resolution convolution kernel.

It is critically important that the CT datasets are analyzed by a cardiovascular imaging specialist with sufficient experience in cardiac CT. Image interpretation should always include the recording of overall image quality based on a multi-item Likert scale, ranging from nondiagnostic to excellent. In addition, we recommend that the evaluability of the coronary arteries be assessed using the 15-segment model or similar models [44]. Non-evaluability of a mid or distal coronary artery segment may not be relevant, depending on the vessel size and coronary artery distribution type. However, high diagnostic image quality of at least the proximal coronary arteries is crucial. If severe artifacts impede proximal coronary artery assessment, censoring the segment as diseased and referral for ICA is most often the consequence. Consistent with the current literature, coronary artery stenosis ≥70% of the proximal coronary artery segment and/or stenosis ≥50% of the left main coronary artery (CAD–RADS category ≥ 4) on CTA is considered to indicate the presence of relevant CAD in the setting of patients evaluated for TAVI [4,45].

## 6. Overview of the Current Literature

### 6.1. CTA Alone for Coronary Artery Assessment and TAVI Evaluation

The utilization of CTA for coronary artery assessment in patients scheduled for TAVI has been investigated previously [7,8,9,10,11,12,13,14,15,16,17,18,19,20,21,22,23]. An overview of available study experience in the context of combined coronary CTA and TAVI evaluation is provided in Table 1. In the following section we will present data of a subset of these studies.

Pontone et al. [7] provided the first systematic insights on the value of CTA for assessing the size of the aortic annulus, the peripheral access site conditions, and the presence of significant coronary artery stenosis in 60 patients. Based on their experiences with a 64-slice system, they concluded that comprehensive CT evaluation of patients referred for TAVI is feasible, allows for a more accurate assessment of the aortic annulus than transesophageal echocardiography, and may replace peripheral angiography in all patients and ICA in patients without relevant CAD. The patient-based analysis showed a sensitivity, specificity, positive predictive value (PPV), and negative predictive value (NPV) of 88.5%, 88.2%, 85.2%, and 90.9%, respectively.

A different approach to assess the role of CTA was chosen by Chieffo et al. [9]. They retrospectively included 491 patients who had undergone CTA and TAVI at their institution. These were divided into two groups: (A) 375 patients with noninvasive evaluation by CTA before TAVI only, and (B) 116 patients with ICA in addition to CTA. The primary study objective was a comparison of the two groups regarding major adverse cardiovascular and cerebrovascular events at 30 days and one year. The authors reported no differences between the groups regarding the defined endpoint. The occurrence of major adverse cardiovascular and cerebrovascular events was comparable between the groups at one year even after multivariable adjustment to reduce the influence of confounding bias. Based on their suggestion, CTA is a safe and effective noninvasive imaging tool for the routine work-up prior to TAVI and may be used as the primary test, with referral for ICA only if required.

Opolski et al. [12] reported a diagnostic performance of CTA prior to TAVI with a sensitivity, specificity, PPV, and NPV of 98%, 37%, 67%, and 94%, respectively. Their study involved 64-slice (first-generation) dual-source CT. A diameter stenosis threshold of ≥50% was applied for the interpretation of CTA to define relevant CAD.

Annoni et al. [15] used a 256-slice CT system for the detection of significant CAD in patients evaluated for TAVI. CAD was defined as ≥50% diameter stenosis on CTA. In an additional evaluation, ≥70% diameter stenosis was defined to indicate significant CAD. The study used quantitative ICA as the reference standard. The use of invasive FFR was not reported. Patients with previous interventional or surgical myocardial revascularization were also included. Preprocedural CTA performed well when evaluating coronary artery bypass graft patency, whereas stents were evaluable with only moderate success. For the threshold of 70% diameter stenosis, the authors report a patient-based sensitivity, specificity, PPV, NPV, and accuracy of 88%, 91%, 66%, 97%, and 91%, respectively.

The article by Schicchi et al. [18] reported the findings from their prospective single-center study of a total of 223 patients. All patients had severe AS and underwent prospective ECG-triggered high-pitch CTA by a third-generation dual-source system prior to TAVI. ICA served as the reference standard. In the assessment of the CTA diameter stenosis threshold ≥70%, the patient-based analysis revealed sensitivity, specificity, PPV, NPV, and accuracy of 92.5%, 85.8%, 58.7%, 98.1%, and 87.0%, respectively.

Van den Boogert et al. [21] evaluated the ability of CTA prior to TAVI to assess CAD using a contemporary approach. As recommended in the current guidelines for revascularization before TAVI [4], only coronary artery stenosis of the left main or the proximal coronary artery segments was defined as significant CAD. The authors used pooled data from studies with various CT systems and did not focus on technological differences. They found that CTA has the potential to safely obviate the referral for ICA in 52% or 70% of TAVI patients, based on the diameter stenosis threshold used to define relevant CAD (≥50% or ≥70%).

A study from our group compared the diagnostic performance of first-generation with that of third-generation dual-source CT for coronary artery assessment during TAVI evaluation in 192 consecutive patients [23]. Importantly, all patients with known CAD and severe chronic kidney disease were excluded. A diameter stenosis of ≥70% was defined to indicate relevant CAD. Compared with first-generation CT, third-generation dual-source CT allowed for less contrast medium and radiation dose, provided better image quality, and improved diagnostic performance. On a per-patient basis, accuracy (72.9% vs. 91.7%), specificity (59.7% vs. 88.3%), PPV (61.0% vs. 83.3%), and NPV (91.9% vs. 98.2%) for detecting CAD per patient were significantly better (*p* < 0.05 for all) using third-generation dual-source CT, while sensitivity was similar (92.3% vs. 97.2%, *p* = 0.129). Based on these data, an estimated 35% and 55% of the patients analyzed could have safely forgone ICA by use of first- and third-generation dual-source CT, respectively.

### 6.2. Study Experience with CTA-Based FFR Prior to TAVI

An increasing number of studies are focusing on the utility of CT–FFR in the context of CTA evaluation before TAVI [24,25,26,27,28]. Table 2 provides an overview of relevant research articles with this focus. 

Michail et al. [24] published the first prospective study to date using the only commercially available CT–FFR approach for the detection of hemodynamically relevant CAD in 39 selected patients with severe AS and an indication for TAVI. Invasive FFR served as the reference standard in all patients, as ≥30% visual stenosis in at least one coronary artery identified at the time of ICA was an inclusion criterion. Based on the per-patient analysis, sensitivity, specificity, PPV, NPV, and accuracy were 76.5%, 77.3%, 72.2%, 81.0%, and 76.9%, respectively.

In a recent retrospective study, Gohmann et al. [25] used ML–CT–FFR to assess for CAD in 216 patients indicated for TAVI. On a per-patient basis, the calculated sensitivity, specificity, PPV, and NPV were 76.9%, 64.5%, 34.0%, and 92.1%, respectively. ML–CT–FFR was shown to further reduce the need for ICA in comparison with CTA despite the challenges faced in the population of patients scheduled for TAVI. The proportion of patients with exclusion of relevant CAD was approximately 44% based on the authors’ analysis.

Brandt et al. [26] evaluated 95 consecutive patients retrospectively. In all patients, preprocedural TAVI-CT and quantitative analysis of ICA were available. Indication for ICA was CAD–RADS ≥ 4, or ML–CT–FFR ≤ 0.8. The diagnostic performance on a per-patient basis using this approach, in terms of sensitivity, specificity, PPV, and NPV, was 100%, 78%, 40%, and 100%, respectively. This led to the conclusion that the combination of CT–FFR and the CAD–RADS classification system for coronary artery stenosis holds the potential to significantly reduce the number of referrals for ICA by up to 68%.

Aquino et al. [27] reported clinical outcomes from their retrospective study on the role of CT–FFR in pre-TAVI work-up. In total, 196 patients were included. A subset of 119 included patients ultimately underwent TAVI. The median follow-up time was 18 months. Major adverse cardiac events (including nonfatal myocardial infarction, unstable angina, cardiac death, and hospital admission due to heart failure) occurred in 16%. CT–FFR ≤ 0.75 was independently associated with major adverse events when adjusting for potential confounders. Furthermore, CT–FFR significantly improved the predictive value of CTA prior to TAVI.

Most recently, Peper et al. [28] retrospectively analyzed 338 patients who had undergone CTA with CT–FFR before TAVI. Quantitative analysis of ICA served as the reference standard. The main exclusion criteria were previous myocardial revascularization (surgical or interventional) and nonapplicable CT–FFR, for example due to insufficient image quality. The reported sensitivity, specificity, PPV, and NPV were 76.9%, 64.5%, 34.0%, and 92.1% for CTA alone vs. 84.6%, 88.3%, 63.2%, and 96.0% for CT–FFR, respectively. Based on the area under the receiver operating characteristics curve, the diagnostic performance of CT–FFR was significantly improved in comparison with CTA alone. The authors concluded that a CT–FFR-guided work-up of AS patients prior to TAVI could avoid ICA in as many as 57.1%.

### 6.3. Available Meta-Analyses on CTA for Coronary Artery Evaluation before TAVI

There are three meta-analyses available that focus on the performance of CTA in diagnosing CAD in the setting of the preprocedural TAVI period [46,47,48]. The publication by Chaikriangkrai et al. [46] included 13 studies with a total of 1498 patients. However, not all underwent TAVI: altogether, 26% of the patients were referred to surgical aortic valve replacement instead. The meta-analyses by van den Boogert et al. [47] and Chaikriangkrai et al. [46] were published in 2018, but the most recent one by Gatti et al. [48] was published in 2022. The latter compiled data from 14 studies with a total of 2533 patients and applied a bivariate random-effects model to summarize the diagnostic performance of CTA for coronary artery assessment before TAVI. They present a pooled per-patient sensitivity and specificity of 97% and 68%, respectively. Although one publication on CT–FFR was included, this was not the main focus of the contribution. Based on their data, the authors show that CTA has an excellent diagnostic accuracy for assessing significant CAD in patients evaluated for TAVI. Overall, they reported that systematic coronary artery assessment with CTA could avert more than 40% of the ICA procedures. This meta-analysis separately evaluated the diagnostic performance of single-heartbeat CT systems and found that these contemporary CT systems may have a significantly better specificity of 82% compared with a specificity of 60% using other CT technology (*p* < 0.0001) for the assessment of significant CAD in patients referred for TAVI. This effect is most likely due to the high temporal resolution and emphasizes the recommendation to use the most current CT technology available for this challenging population of patients.

## 7. Discussion

Based on the progressive demographic changes in industrialized countries, advancements of the TAVI procedure, and extension of its indication, it is expected that the number of patients with AS who are referred for an interventional treatment will further increase. Since the majority of patients treated with TAVI are characterized as elderly and frail, the invasiveness of the procedure has been reduced significantly over the past two decades while the level of safety has been maintained or even elevated. As an example, the use of local instead of general anesthesia or the reduction of femoral sheath diameters can be noted as important steps in the evolution of TAVI. Another concept aimed at streamlining TAVI and minimizing its invasiveness is SLIM (single-arterial access and low contrast medium usage) [49]; it was shown that this approach decreased the rates of acute kidney injury as well as access-site complications and led to a reduction in the length of the hospital stay. At the same time, procedural safety and success were not compromised. The trend towards minimizing invasiveness of TAVI also includes the preprocedural work-up. During the early years of TAVI, transesophageal echocardiography was performed in every patient to verify the severity of AS and to permit prosthesis sizing. Meanwhile, CT has been shown to provide more accurate and noninvasive imaging of the aortic annulus and beyond. It has therefore become the central evaluation tool in the context of TAVI, and transesophageal echocardiography is now used only as an adjunctive diagnostic method in selective cases. If pre-TAVI CTA was thoroughly proven to reliably rule out relevant CAD and to obviate the need for further downstream testing in at least a subset of the complex cohort of patients scheduled for TAVI, it could serve as a “one-stop shop” in these cases. There are a number of possible complications related to ICA that can have serious consequences [50], which reinforces the interest in a noninvasive diagnostic method. CTA has the potential to provide information on all relevant aspects of the procedure. However, there are certain considerations and prerequisites for coronary artery evaluation in the setting of CTA prior to TAVI. Specifically, it is advisable to aim for balanced fluid management depending on the patients’ renal function and acuteness of admittance, to use the most contemporary CT scanner generation available in conjunction with a dedicated CT protocol, and to assess image data involving cardiovascular imaging specialists ideally integrating knowledge of clinical cardiology and expertise in cardiac CT.

As mentioned above, retrospective ECG-gating presents an advantageous acquisition technique for combined coronary CTA and TAVI planning because the choice of the transcatheter heart valve can be made based on the largest dimension throughout the cardiac cycle and coronary artery assessment is possible in the most suitable cardiac phase. However, retrospective ECG-gating results in the highest possible radiation dose. This may arguably be without clinical consequence in elderly patients; however, the mean age and surgical risk profile of patients treated with TAVI have decreased steadily and significantly over time. When considering increasingly younger TAVI populations, CTA could be even more advantageous due to lower cardiovascular risk profiles associated with less atherosclerotic burden. At the same time, it is good practice to keep the radiation dose as low as reasonably possible, although there is no evidence that radiation from medical imaging is associated with any cancer risk in an adult population [51].

If imaging of the aortic annulus is inaccurate, prosthesis oversizing may occur. Oversizing above a certain degree has been shown to increase the risk of annular rupture [52], a rare but serious TAVI-associated complication. Image salvage for correct annulus sizing and accurate coronary artery assessment, despite the presence of image artifacts, is less likely when prospective ECG-triggering is used. The consequence may be repeated CTA examinations or unnecessary referrals for ICA, which entails additional radiation dose and contrast material or, in case of ICA, the additional risks inherent to an invasive procedure.

## 8. Conclusions

Coronary artery assessment during routine preprocedural CTA for TAVI planning seems feasible in a significant proportion of patients with severe symptomatic AS using contemporary CT technology and a retrospective ECG-gating technique. According to the available literature, CTA is a promising means to noninvasively assess the coronary arteries in the evaluation of selected patients for TAVI to aid clinical decision-making regarding whether invasive testing is required. Utilization of CTA-based FFR may be of additional value for this purpose.

## 9. Future Directions

Continuing research is required to corroborate the findings of previous, predominantly retrospective studies within this area of interest. To establish the role of CTA as a safe and effective first-line screening test for CAD in the preprocedural setting of TAVI, carefully designed prospective randomized trials are required. Also, the usefulness of a diagnostic algorithm to preselect cases that are appropriate for a primarily noninvasive CT examination and those that should be referred for ICA in cases of suspected relevant CAD deserves further investigation. Figure 2 proposes an algorithm with CTA as the central diagnostic tool involving ICA only if suspicion of relevant CAD is present.

Ideally, coronary artery assessment should be performed in every patient undergoing CTA prior to TAVI. However, some of the research articles that focused on CTA for coronary artery assessment and TAVI evaluation excluded certain patients in their analyses, such as those with severe renal insufficiency and known CAD with or without previous revascularization. Patients with these characteristics can be assigned to the highest risk category. Hence, if the institutional capacity for thorough coronary artery evaluation by CTA is not available for all patients undergoing TAVI, these exclusion criteria together with coronary Agatston score cutoffs may facilitate the preselection of patients with a lower probability of relevant coronary artery stenosis in whom relevant CAD is likely to be ruled out by pre-TAVI CTA.

Future studies should also involve CT–FFR due to its suggested incremental value for diagnosing relevant CAD. An unresolved issue is that the commercially available CT–FFR algorithm requires out-of-hospital transfer of the CT dataset and extensive calculation time. There are point-of care approaches to on-site CT–FFR. These are based on computational fluid dynamics or virtually instantaneous ML and have been studied with promising results. However, all of the on-site CT–FFR applications are research prototype algorithms that are currently not commercially available.

## Figures and Tables

**Figure 1 diagnostics-13-01327-f001:**
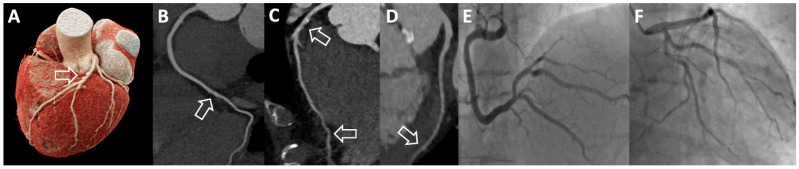
Evaluation of an 81-year-old male patient with severe AS prior to TAVI. Retrospective ECG gating was used as the CT acquisition technique and provided excellent image quality. As shown by the three-dimensional volume rendering (**A**), as well as the vessel views of the dominant right coronary artery (**B**), left anterior descending coronary artery (**C**) and left circumflex coronary artery (**D**), only minimal stenosis (1–24% luminal diameter stenosis; CAD–RADS 1; white arrows) was found. ICA of the right (**E**) and left coronary artery (**F**) was redundant and confirmed absence of relevant CAD.

**Figure 2 diagnostics-13-01327-f002:**
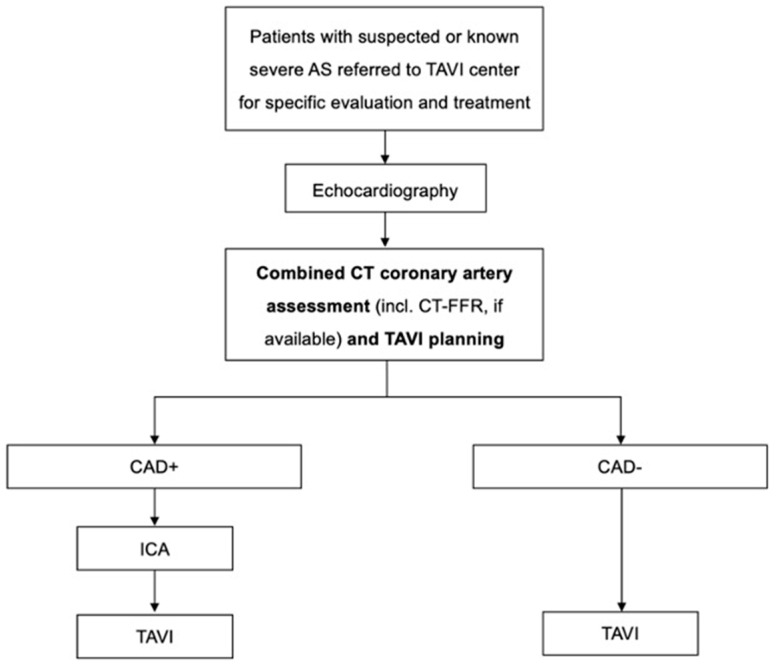
Proposed diagnostic algorithm of combined CT coronary artery assessment and TAVI planning. Abbreviations: CAD+ = suspicion of relevant coronary artery disease based on computed tomography; CAD− = relevant coronary artery disease ruled-out by computed tomography; CT–FFR = fractional flow reserve from computed tomography angiography; ICA = invasive coronary angiography; TAVI = transcatheter aortic valve implantation.

**Table 1 diagnostics-13-01327-t001:** Relevant studies focusing on the diagnostic performance of preprocedural CTA before TAVI for the detection of relevant CAD.

Study	Year of Publication	Design	Number of Patients (n)	Age (Years)	Male (%)	Reference Standard	CT System	CT Evaluability (%)	CT Definition of Relevant CAD (% Stenosis)	CT–FFR	Sensitivity (%) ^$^	Specificity (%) ^$^	PPV (%) ^$^	NPV (%) ^$^
Pontone et al. [7]	2011	R, S	60	80 ± 8	36.7	ICA	64-slice	86.7 (patients)	≥50	No	88.5	88.2	85.2	90.9
Andreini et al. [8]	2014	R, S	325	81.1 ± 6.6	40.6	ICA	64-slice	74.8 (patients)	≥50	No	89.7	90.8	80.6	95.4
Chieffo et al. [9]	2015	R, S	491	79.9 ± 7.8 *	46.7 *	N/A **	64-slice	95.7 (patients)	≥50	No	N/A **	N/A **	N/A **	N/A **
Hamdan et al. [10]	2015	R, S	115	79.8 ± 7.2 *	34.8 *	ICA	256-slice	93.6 (segments) ^†^	≥50	No	95.9	72.7	72.3	96.0
Harris et al. [11]	2015	R, S	100	79.6 ± 9.9	61.0	ICA	128-slice DS	94.7 (vessels)	≥50	No	98.6	55.6	85.7	93.8
Opolski et al. [12]	2015	R, S	475	82 ± 6	40.8	ICA	64-slice DS	52.0 (patients)	≥50	No	98.1	37.1	67.3	93.8
Matsumoto et al. [13]	2016	R, S	60	84.4 ± 4.6	28.3	ICA	320-slice	78.3 (patients)	≥50	No	91.7	58.3	59.5	91.3
Rossi et al. [14]	2017	R, S	140	82.3 ± 7.7	48.6	QCA	128-slice DS	96.6 (vessels)	≥50% and ≥70%	No	78.3 ^§^	73.5 ^§^	36.7 ^§^	94.5 ^§^
Annoni et al. [15]	2018	R, S	115	82.5 ± 6.2	55.7	ICA	256-slice	95.3 (segments)	≥50 and ≥70	No	88 ^§^	91 ^§^	66 ^§^	97 ^§^
Hachulla et al. [16]	2019	R, S	84	83.0 ± 6.8	52.4	ICA	128-slice DS	89.3 (patients)	≥70	No	100	86	87	100
Strong et al. [17]	2019	R, S	200	83.4 ± 5.9	40.0	ICA	64-slice DS	42.5 (patients)	≥50	No	100.0	42.0	47.6	100.0
Schicchi et al. [18]	2019	P, S	223	79.2 ± 4.9	N/A	ICA	192-slice DS	95.0 (segments)	≥50 and ≥70	No	92.5 ^§^	85.8 ^§^	58.7 ^§^	98.1 ^§^
Shuai et al. [19]	2021	R, S	130	73.3 ±6.4	47.1	ICA	256-slice	N/A	≥50	No	96	89	71	98
Meier et al. [20]	2021	R, M	127	82.3 ± 7.3	38.6	ICA	64- and 256-slice	76.3 (vessels)	≥50 and ≥70 ^§^	No	42.8	97.8	56.3	96.3
van den Boogert et al. [21]	2022	R, M	1060	81.7 ± 6.6	42.7	ICA	Various (at least 64-slice)	80.3 (proximal segments)	≥50 and ≥70 ^§^	No	96.7	87.5	66.9	99.0
Malebranche et al. [22]	2022	R, S	100	82.3 ± 6.5	30.0	QCA	128-slice DS	16.0 (patients)	≥50 and ≥70 ^§^	No	100.0	11.4	32.6	100.0
Renker et al. [23]	2022 ^#^	R, S	192	80.0	36.5	ICA + invasive FFR	64- vs. 192-slice DS	96.5 (vessels)	≥70	No	97.2 *	88.3 *	83.3 *	98.2 *

Abbreviations: CAD = coronary artery disease; CT–FFR = fractional flow reserve from computed tomography angiography; DS = dual-source CT; ICA = invasive coronary angiography; M = multicenter; NPV = negative predictive value; P = prospective; PPV = positive predictive value; QCA = quantitative analysis of invasive coronary angiography; R = retrospective; S = single-center. ^$^ On a per-patient basis, including noninterpretable vessels/examinations. ^†^ Interpretable were 93.6% segments in the 92 patients without prior bypass surgery and 95.2% grafts in the patients with prior bypass surgery. * Based on data provided for the experimental study group. ** Comparison of the rate of major adverse cardiovascular and cerebrovascular events in group A (CTA only, no ICA) and group B (CTA + ICA). ^§^ Results for diagnostic performance of CT are based on ≥70% diameter stenosis threshold. ^#^ Year of acceptance for publication.

**Table 2 diagnostics-13-01327-t002:** Relevant studies focusing on the diagnostic performance of preprocedural CT–FFR before TAVI for the detection of relevant CAD.

Study	Year of Publication	Design	Number of Patients (n)	Age (years)	Male (%)	Reference Standard	CT System	CT–FFR Feasibility (%)	Definition of Relevant CAD by CT	CT–FFR Approach	Sensitivity (%) ^$^	Specificity (%) ^$^	PPV (%) ^$^	NPV (%) ^$^
Michail et al. [24]	2021	P, S	39	76.2 ± 6.7	71.8	Invasive FFR	320-slice	92.3 (patients)	CT–FFR ≤ 0.80	CFD-based, external data transfer	76.5	77.3	72.2	81.0
Gohmann et al. [25]	2022	R, S	216	84.4 ± 6.2	58.3	QCA	128-slice DS	79.4 (patients)	CT–FFR ≤ 0.80	ML-based, on-site *	93.6	58.3	52.3	94.9
Brandt et al. [26]	2022	R, S	95	78.6 ± 8.8	47	ICA	128-slice DS	83.6 (patients)	CAD–RADS ≥ 4, or CT–FFR ≤ 0.80	ML-based, on-site *	100	78	40	100
Aquino et al. [27]	2022	R, S	196 ^§^	75 ± 11	43.9	N/A **	192-slice DS	89.9 (patients) ^†^	CT–FFR ≤ 0.75	ML-based, on-site *	N/A **	N/A **	N/A **	N/A **
Peper et al. [28]	2022	R, S	338	81.0 ± 6.5	42.3	QCA + invasive FFR	64-slice or DS or 256-slice	88.7 (patients)	CT–FFR ≤ 0.80	CFD-based, on-site *	84.6	88.3	63.2	96.0

Abbreviations: CAD = coronary artery disease; CFD = computational fluid dynamics; CT–FFR = fractional flow reserve from computed tomography angiography; ICA = invasive coronary angiography; ML = machine learning; NPV = negative predictive value; P = prospective; PPV = positive predictive value; QCA = quantitative analysis of invasive coronary angiography; R = retrospective; S = single-center; TAVI = transcatheter aortic valve implantation. ^$^ On a per-patient basis, including only patients with feasible CT–FFR calculation. ^†^ Assessment of both, CAD–RADS and CT–FFR was possible in 83.6% of the included patients. * Research prototype, not available for commercial use. ^§^ Thereof, 119 patients underwent TAVI. ** Comparison of the rate of major adverse cardiovascular events in patients with CT–FFR ≤ 0.75 and >0.75.

## Data Availability

No new data were created or analyzed in this study. Data sharing is not applicable to this article.

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
