# Peer review of "Combined CT Coronary Artery Assessment and TAVI Planning"

_diagnostics, 2023, doi:10.3390/diagnostics13071327_

Round 1

Reviewer 1 Report

Overall this is a high quality review on the topic.

Only minor comments:

In the introduction, authors state that as many as approximately 80% of the patients 36 evaluated for transcatheter aortic valve implantation (TAVI) have coexistent CAD.”

This corresponds to the higher rate reported in selected and high risk population, using this data transmit a biased information in my opinion, please offer the range of published prevalence.

I miss data on the percentage of interpretable CTs for this purpose (i.e. coronary artery evaluation). In the same line of thought, it is not clear if the sensitivity, specificity, PPV and NPV in the mentioned studies are calculated based on the whole set of CTs or on the interpretable CTs.

Discussion may be improved:

Based on the data presented and their expertise, authors should discuss optimal patient selection for CT coronary assessment (every patient, high risk patient, low risk patient…). Authors may continue their sentence “However, there are certain considerations and prerequisites for coronary artery evaluation in the setting of CTA prior to TAVI”.

What is authors opinion on CT-FFR? Should It be routinely included if possible?

If authors feel comfortable with the idea, a figure with a proposed algorithm based on the presented evidence may be added

Reviewer 2 Report

Renker et al. conducted an interesting review on the currently available evidence for the use of CTA for the diagnosis of CAD in patients scheduled for TAVI. The manuscript is well written. Please find some suggestions below:

Could you add data on calcium scoring alone to rule out significant CAD? How many patients would not require ICA based on non-contrast calcium scoring alone, since this method is available in all TAVI-centers.

Most centers still rely on ICA as standard of care. Given the current limitations of CTA-based FFR and ML-FFR solutions, what would the authors suggest as a standard of care algorithm for the workup of CAD in AS-patients using the assumingly widely available CT-technique?

In most cases of significant CAD in patients scheduled for TAVI one would assume a stable state of the disease and revascularization might not be beneficial, even if significant CAD is present. I would suggest adding a paragraph on the current evidence for revascularization before TAVI and its limitations (especially the ACTIVATION study could be used as an argument to include functional lesion assessment and not only rely on anatomic evaluation of CAD) and you could give an outlook on the awaited studies (NOTION 3 and COMPLETE TAVR)

Minor:

Please decide on using pre-hydration or prehydration and use it consequently.

Please delete “with” in line 183

Please rephrase line 119: …, especially in cases with fibrosis as main contributor to AS [39].

Reviewer 3 Report

This is a journal that aims to review addresses technological prerequisites as well as CT protocol considerations, considers pitfalls, reviews the current literature regarding an integrated coronary artery assessment during routine preprocedural CTA for TAVI, and provides an  overview of unresolved issues and perspectives for future research within this field of interest. This is an interesting topic. However, there are some issues that need to be addressed.

1. 2. Technological Prerequisites - 4. CT Protocol. Technological Prerequisites and technical methods occupy too much space when looking at the entire paper. Modification is needed to focus on the interpretation and utilization of the results rather than the technical method.

2. 6.0 Overview of the Current Literature. In Sections 6.1 and 6.2, it is repetitive and inappropriate to refer to all the findings in Tables 1 and 2. It is recommended to summarize overlapping results to help readers understand. .

3. 7.0 Discussion. The authors wrote that “"If pre-TAVI CTA could reliably rule out relevant CAD and obviate the need for further downstream testing in at least a subset of patients scheduled for TAVI, it would serve as a “one-stop shop”. However, the high NPV value of CTA is not a new characteristic, and the ability to screen patients without CAD through CTA has already been sufficiently verified in patients without severe AS.
Tables 1 and 2 show that the PPV of CTA ranges from 30% to 80%, which means that many patients undergo ICA unnecessarily with CTA. In order for this review to have more meaning, I think there should be a technical method that can improve the PPV of CTA and its effect.

Round 2

Reviewer 3 Report

This is a journal that aims to review addresses technological prerequisites as well as CT protocol considerations, considers pitfalls, reviews the current literature regarding an integrated coronary artery assessment during routine preprocedural CTA for TAVI, and provides an overview of unresolved issues and perspectives for future research within this field of interest. 

This paper well summarized the technical aspects of CT and explained the role of CT in TAVR.